# Emergence of Third-Generation Cephalosporin-Resistant *Morganella morganii* in a Captive Breeding Dolphin in South Korea

**DOI:** 10.3390/ani10112052

**Published:** 2020-11-06

**Authors:** Seon Young Park, Kyunglee Lee, Yuna Cho, Se Ra Lim, Hyemin Kwon, Jee Eun Han, Ji Hyung Kim

**Affiliations:** 1Infectious Disease Research Center, Korea Research Institute of Bioscience and Biotechnology, Daejeon 34141, Korea; lovesun139@kribb.re.kr (S.Y.P.); dlatpfk13@kribb.re.kr (S.R.L.); hena0922@kribb.re.kr (H.K.); 2Division of Animal and Dairy Sciences, College of Agriculture and Life Science, Chungnam National University, Daejeon 34134, Korea; 3Cetacean Research Institute, National Institute of Fisheries Science, Ulsan 44780, Korea; moby19@korea.kr (K.L.); tnvlfldj@gmail.com (Y.C.); 4Laboratory of Aquatic Biomedicine, College of Veterinary Medicine, Kyungpook National University, Daegu 41566, Korea

**Keywords:** marine mammal health, cetacean, antimicrobial resistance, ampR, zoonotic infection

## Abstract

**Simple Summary:**

The emergence of antimicrobial resistance (AMR) has become an important consideration in animal health, including marine mammals, and several potential zoonotic AMR bacterial strains have been isolated from wild cetacean species. Although the emergence of AMR bacteria can be assumed to be much more plausible in captive than in free-ranging cetaceans owing to their frequent contact with humans and antibiotic treatments, the spread and its impacts of AMR bacteria in captive animals have not been adequately investigated yet. Here in this study, we present evidence on the presence of multidrug-resistant potential zoonotic bacteria which caused fatal infection in a captive dolphin bred at a dolphinarium in South Korea.

**Abstract:**

The emergence of antimicrobial resistant (AMR) strains of *Morganella morganii* is increasingly being recognized. Recently, we reported a fatal *M. morganii* infection in a captive bottlenose dolphin (*Tursiops truncatus*) bred at a dolphinarium in South Korea. According to our subsequent investigations, the isolated *M. morganii* strain KC-Tt-01 exhibited extensive resistance to third-generation cephalosporins which have not been reported in animals. Therefore, in the present study, the genome of strain KC-Tt-01 was sequenced, and putative virulence and AMR genes were investigated. The strain had virulence and AMR genes similar to those of other *M. morganii* strains, including a strain that causes human sepsis. An amino-acid substitution detected at the 86th residue (Arg to Cys) of the protein encoded by *ampR* might explain the extended resistance to third-generation cephalosporins. These results indicate that the AMR *M. morganii* strain isolated from the captive dolphin has the potential to cause fatal zoonotic infections with antibiotic treatment failure due to extended drug resistance, and therefore, the management of antibiotic use and monitoring of the emergence of AMR bacteria are urgently needed in captive cetaceans for their health and conservation.

## 1. Introduction

The spread of antimicrobial resistance genes (ARGs) and their acquisition in potential zoonotic bacterial pathogens have been recognized as serious threats to human and animal health [1]. As in other animals, the emergence of antimicrobial resistant (AMR) bacteria in cetaceans has become an important consideration [2]. Several potential zoonotic bacterial species have been isolated from wild cetacean species inhabiting the United States [3] and Brazil [4]. Moreover, the emergence of AMR bacteria can be assumed to be much more plausible in captive than in free-ranging marine mammals owing to their frequent contact with humans and antibiotics used for prophylactic or curative treatment. Indeed, AMR bacteria, including those causing fatal infections, have been reported in captive dolphins bred in a dolphinarium [5,6,7,8]. However, the mechanisms of AMR in captive marine mammals have not been adequately investigated because of limited genetic (or genomic) information. Therefore, there is an urgent need to investigate the genomic characteristics of AMR bacteria isolated from captive cetaceans to evaluate the current status of AMR in dolphinariums and to clarify the potential health risks for marine mammals and the humans they encounter.

*Morganella morganii* is a motile flagellated, straight, rod-shaped, Gram-negative bacterium that is ubiquitous in the environment and intestinal tracts of humans and animals. *M. morganii* has clinical significance as a potential causal pathogen of nosocomial and animal infections [9]. Similar to other Enterobacteriaceae species, *M. morganii* shows intrinsic resistance to β-lactam antibiotics, including first- and second-generation cephalosporins. In addition, the rates of drug resistance and AMR genes in this bacterium have recently increased [9]. Moreover, this bacterium has been implicated in the notorious histamine fish poisoning (or scombroid poisoning) based on the sporadic presence of histidine decarboxylase (*hdc*) [10]. Recently, we reported the occurrence of a fatal *M. morganii* infection in a captive bottlenose dolphin (*Tursiops truncatus*) bred at a dolphinarium in South Korea [11]. In subsequent investigations of the isolated *M. morganii* strain KC-Tt-01, we found that it exhibited intrinsic resistance against several antibiotics and was extensively resistant to third-generation cephalosporins. Therefore, in the present study, we characterized the genome of *M. morganii* strain KC-Tt-01 and its relation to the phenotypic AMR profile.

## 2. Materials and Methods

### 2.1. Ethics Statement

The dolphins mentioned in this work were cared (or managed) in a captive environment and handled according to Korean law (Act on the management of zoos and aquariums, Act 14227/2016); all the samples obtained (in vivo diagnostic swabs and blood, and post-mortem samples) were collected according to the above and within Korean law (Wildlife protection and management act, Act 13882/2016), which establishes the management objectives and prescriptions to maintain the species under human care. The animal study was reviewed and approved by the ethics and welfare committee (Approved number: 2017-Animal Experiment-15) in the National Institute of Fisheries Science, Ministry of Oceans and Fisheries, Republic of Korea.

### 2.2. Origin of Clinical Isolate M. morganii Strain KC-Tt-01

The *M. morganii* strain KC-Tt-01, which caused fatal fibrino-hemorrhagic bronchopneumonia, was originally isolated from the pericardial fluid of a captive female bottlenose dolphin (*T. truncatus*) bred at a dolphinarium in South Korea [11]. Strain KC-Tt-01 was stored in tryptic soy broth (Difco, Detroit, MI, USA) with 10% glycerol at −80 °C until use.

### 2.3. Determination of Phenotypic Antibiotic Resistance of M. morganii Strain KC-Tt-01

Antimicrobial susceptibility of strain KC-Tt-01 was tested using the disk diffusion method according to the guidelines of the Clinical and Laboratory Standards Institute [12]. A total of 13 groups of antimicrobial agents were used: penicillins, β-lactam/β-lactamase inhibitor combinations, cephems (including first-, second-, third-, and fourth-generation cephalosporins), monobactams, carbapenems, aminoglycosides, tetracyclines, fluoroquinolones, quinolones, folate pathway inhibitors, macrolides, phenicols, and polymyxin (Table 1). The minimum inhibitory concentrations (MICs) of the selected antimicrobial agents were determined using MIC evaluator strips (Oxoid Ltd., Basingstoke, UK). For quality control, *Escherichia coli* ATCC25922 and ATCC35218 were used for the analysis.

### 2.4. Sequencing and Analysis of the M. morganii Strain KC-Tt-01 Genome

*M. morganii* strain KC-Tt-01 was cultured overnight on 5% sheep blood agar (Hanil Komed, Seongnam, Korea) at 37 °C. Bacterial genomic DNA was isolated using a DNeasy blood and tissue kit (QIAGEN, Hilden, Germany) following the manufacturer’s protocol. Genome sequencing was carried out by Macrogen Inc. (Seoul, Korea) using a hybrid approach with the PacBio RS II system (Pacific Biosciences, USA) and the HiSeq 2000 system (Illumina, San Diego, CA, USA). The generated sequences (1,242,336,868 bp; 135,696 reads) were assembled using HGAP v.3.0 (https://github.com/PacificBiosciences/Bioinformatics-Training/wiki/HGAP), and the Illumina paired-end reads (965,969,557 bp, 9,572,268 reads) were mapped to the assembled contigs to improve the accuracy of the sequenced genome. Genome annotation was conducted using the National Center of Biotechnology Information Prokaryotic Genome Annotation Pipeline (http://www.ncbi.nlm.nih.gov/books/NBK174280/), and PHASTER (http://phaster.ca/) analysis was used to detect prophages. To assess the genomic relatedness to other *Morganella* species, the average nucleotide identity was analyzed using OrthoANI (http://www.ezbiocloud.net/tools/orthoani). Putative virulence and antimicrobial resistance genes were preliminarily screened by searching against the Virulence Factor (http://www.mgc.ac.cn/VFs/) and ARG-ANNOT (http://en.mediterranee-infection.com/article.php?laref=283&titre=arg-annot-) databases, respectively, and were ultimately identified by manual comparisons with those reported for other *M. morganii* strains in GenBank, including strain KT, which causes human sepsis [15].

### 2.5. Culture Deposition and Nucleotide Sequence Accession No.

*M. morganii* strain KC-Tt-01 was deposited in the Korean Culture Center of Microorganisms (KCCM) as KCCM 90280. The complete genome sequence of the strain has been deposited in GenBank under accession number CP025933.

## 3. Results and Discussion

The antimicrobial susceptibility profile of *M. morganii* strain KC-Tt-01 is shown in Table 1. The strain was resistant to ampicillin (MIC >256 μg/mL), amoxicillin-clavulanate (>256 μg/mL), ampicillin-sulbactam, cephalothin, cephazolin, cefoxitin, cefuroxime, cefotaxime (64 μg/mL), ceftazidime, aztreonam, erythromycin (>256 μg/mL), and polymyxin B. The intrinsic resistance of *M. morganii* against ampicillin, amoxicillin-clavulanate, first- and second-generation cephalosporins, tetracycline, erythromycin, and polymyxin B has been well documented [12,13,14], and similar results were obtained for strain KC-Tt-01 in this study. However, our isolate was also resistant to cefotaxime, a third-generation cephalosporin. Therefore, we sequenced the genome of strain KC-Tt-01 to determine its resistance mechanism and provide genomic insights into this potential zoonotic pathogen infecting marine mammal species.

The fully assembled complete genome of strain KC-Tt-01 was 3,824,890 bp long (G + C content, 51.1%), and plasmids were not detected. The annotated genome included 3,611 genes, 3,506 coding sequences, 22 rRNAs (5S, 16S, and 23S), 79 tRNAs, and four non-coding RNAs. In addition, five prophage regions (three intact and two incomplete) were detected (Appendix A). The KC-Tt-01 genome showed the highest similarity to *M. morganii* strain FDAARGOS_63 (CP026046, 98.94%) based on OrthoANI analysis, and clustered with other *M. morganii* strains in a phylogenetic tree (Figure 1). Most of the potential virulence genes in KC-Tt-01 were very similar to those in strain KT, including the histidine decarboxylase gene cluster (*hdcT1*, *hdc*, *hdcT2*, and *hisRS*); however, repeats-in-toxin (RTX) toxin hemolysin A was detected only in our isolate (Appendix A). Moreover, the genome included genes known to be involved in resistance to β-lactams (*ampD*, *ampH*, *ampR*, *dha-4*, and *emrAB*), phenicols (*catA2*), and polymyxin (*arnA*), which were very similar to those identified in other *M. morganii* strains included in GenBank (Appendix A). These results support the intrinsic resistance of *M. morganii* strain KC-Tt-01 to β-lactam antibiotics, including first- and second-generation cephalosporins and polymyxin B. Although the emergence of third-generation cephalosporin-resistant *M. morganii* has been reported, all previously reported strains originated from nosocomial infections, and TEM β-lactamase production was implicated in the resistance mechanism (e.g., TEM-10 and TEM-21) [16,17]. However, no such β-lactamase was found in strain KC-Tt-01, whereas we detected a substitution at the 86th amino acid (Arg to Cys) of the protein encoded by *ampR* only in strain KC-Tt-01. This alteration might be associated with the extended resistance to third-generation cephalosporins, as has been reported in other Enterobacteriaceae species [18].

Although the origin and route of infection of strain KC-Tt-01 in a captive dolphin bred at the dolphinarium remain unclear, these findings strongly suggest that the captive dolphin-isolated *M. morganii* may have the potential to cause fatal zoonotic infections in humans considering the similarities in their virulence genes, along with failure in antibiotic treatment due to the prolonged cephalosporin resistance. Unfortunately, only a few studies on the AMR of bacteria in dolphinariums and resident dolphins have been conducted to date [5,6,7,8], making direct comparisons difficult. Nevertheless, the acquisition of AMR clearly poses a public health risk for humans who come in direct contact with dolphins (e.g., dolphin trainers and veterinarians). Moreover, captive dolphins harboring AMR bacteria could be a source of unintentional and unexpected spread of ARGs and AMR bacteria in wild dolphins and marine environments during their rehabilitation. Therefore, the management of antibiotic use and monitoring of the emergence of ARGs and AMR bacteria are urgently needed in captive cetaceans, at least during their rehabilitation programs, for their health and conservations. To the best of our knowledge, this is the first report of the occurrence of a third-generation cephalosporin-resistant *M. morganii* infection in animals including marine mammals and its associated genomic characteristics.

## Figures and Tables

**Figure 1 animals-10-02052-f001:**
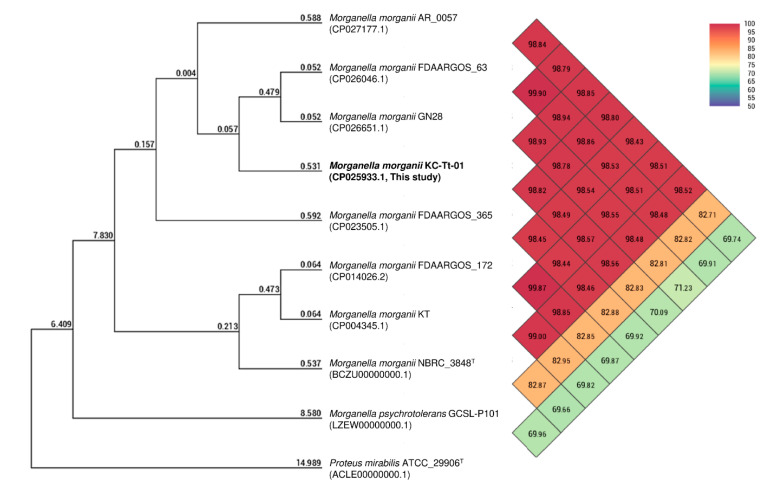
Phylogenetic trees based on the orthologous average nucleotide identity (orthoANI) values calculated using available genomes of *Morganella morganii*, *M. psychrotolerans*, and *Proteus mirabilis*. Comparisons between two strains are given at the junction point of the diagonals departing from each strain; i.e., the OrthoANI value for *M. morganii* strain KC-Tt-01 (CP025933.1) and strain GN28 (CP026651.1) is 98.93%. (Two-column fitting image).

**Table 1 animals-10-02052-t001:** Antibiotic resistance profile of *Morganella morganii* strain KC-Tt-01 ^†^.

Test Group	Antimicrobial Agent	Disk Diffusion		MIC (μg/mL)
		Disk Content (μg)	Result	
Penicillins				
	Ampicillin ^‡^	10	R	>256 (R)
β-lactam/β-lactamase inhibitor combinations				
	Amoxicillin	10	R	>256 (R)
	Amoxicillin-clavulanate ^‡^	20/10	R	>256 (R)
	Ampicillin-sulbactam	10/10	R	ND ^§^
	Piperacillin-tazobactam	100/10	S	ND
Cephems (including cephalosporins 1st, 2nd, 3rd, and 4th)				
1st ^‡^	Cephalothin	30	R	ND
	Cephazolin	30	R	ND
2nd ^‡^	Cefoxitin	30	R	ND
	Cefuroxime	30	R	ND
3rd	Cefotaxime	30	R	64 (R)
	Ceftazidime	30	R	ND
4th	Cefepime	30	S	ND
Monobactams				
	Aztreonam	30	R	ND
Carbapenems				
	Imipenem ^¶^	10	I	16
	Meropenem	10	S	0.12
Aminoglycosides				
	Gentamicin	10	S	1
	Amikacin	30	S	2
Tetracyclines				
	Tetracycline ^‡^	30	S	4
Fluoroquinolones				
	Ciprofloxacin	5	S	0.008
	Levofloxacin	5	S	0.12
	Enrofloxacin	5	S	ND
Quinolones				
	Nalidixic acid	30	S	ND
Folate pathway inhibitors				
	Trimethoprim- sulfamethoxazole	1.25/23.75	S	ND
Macrolides				
	Erythromycin	15	R	>256 (R)
Phenicols				
	Chloramphenicol	30	S	ND
Polymyxin				
	Polymyxin B ^‡^	300	R	ND

^†^ The antimicrobial resistance of strain KC-Tt-01 was quantitatively tested and interpreted according to the Clinical and Laboratory Standards Institute (CLSI) guidelines [12]. ^‡^ The intrinsic antimicrobial resistance (AMR) for *Morganella morganii* has been documented [12,13,14]. ^§^ Not done. ^¶^ Based on the CLSI guidelines [12], *Proteus* species, *Providencia* species, and *Morganella* species may have elevated minimal inhibitory concentrations to imipenem via mechanisms other than the production of carbapenemases.

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
