# Peer review of "Emergence of Third-Generation Cephalosporin-Resistant Morganella morganii in a Captive Breeding Dolphin in South Korea"

_animals, 2020, doi:10.3390/ani10112052_

Round 1

Reviewer 1 Report

Overall, the paper showed the antimicrobial resistance characterization of Morganella morganii isolated from dolphin. In particular, since M. morganii is considered to be a pathogen in mammals and humans, studies on its genome and antibiotic resistance can be used as important data to cope with future infectious diseases.This paper is considered suitable for 'Animals' journal.

Minor comments

line 87 and 103: M. morganii -> itelic
line 157 : Enterobacteriaceae -> itelic

Author Response

Overall, the paper showed the antimicrobial resistance characterization of Morganella morganii isolated from dolphin. In particular, since M. morganii is considered to be a pathogen in mammals and humans, studies on its genome and antibiotic resistance can be used as important data to cope with future infectious diseases.This paper is considered suitable for 'Animals' journal.

Response: We greatly appreciate the positive response and helpful advice from reviewer #1. Based on these comments, we have revised our manuscript.

Minor comments

line 87 and 103: M. morganii -> itelic
line 157 : Enterobacteriaceae -> itelic

Response: Thank you for the helpful comments. We have revised the manuscript accordingly.

Reviewer 2 Report

In the opinion of this reviewer, his manuscript did not sufficiently meet these criteria: excellent science, interest to a wide audience, and novelty, so it was unfortunately rejected.

Author Response

Thank you.

Reviewer 3 Report

M. morgania sticks belong to microorganisms of little invasiveness. They rarely cause infections in healthy people, but can cause opportunistic nosocomial infections with severe course and high mortality. Among patients
Outpatient infections of Morganella sp. aetiology are sporadic. However, such infections are possible, therefore the presented observations are a cause for further research on drug-sensitive strains of M. morganella. I recommend accepting the work for printing without corrections.

Author Response

M. morganii sticks belong to microorganisms of little invasiveness. They rarely cause infections in healthy people, but can cause opportunistic nosocomial infections with severe course and high mortality. Among patients
Outpatient infections of Morganella sp. aetiology are sporadic. However, such infections are possible, therefore the presented observations are a cause for further research on drug-sensitive strains of M. morganella. I recommend accepting the work for printing without corrections.

Response: We greatly appreciate the positive response from reviewer #3.